# Multiple microbial guilds mediate soil methane cycling along a wetland salinity gradient

Wyatt H. Hartman,[1] Clifton P. Bueno de Mesquita,[1] Susanna M. Theroux,[1] Connor Morgan-Lang,[2] Dennis D. Baldocchi,[3] Susannah G. Tringe[1,4]

**ABSTRACT**  Estuarine wetlands harbor considerable carbon stocks, but rising sea levels could affect their ability to sequester soil carbon as well as their potential to emit methane ($CH_4$). While sulfate loading from seawater intrusion may reduce $CH_4$ production due to the higher energy yield of microbial sulfate reduction, existing studies suggest other factors are likely at play. Our study of 11 wetland complexes spanning a natural salinity and productivity gradient across the San Francisco Bay and Delta found that while $CH_4$ fluxes generally declined with salinity, they were highest in oligohaline wetlands (ca. 3-ppt salinity). Methanogens and methanogenesis genes were weakly correlated with $CH_4$ fluxes but alone did not explain the highest rates observed. Taxonomic and functional gene data suggested that other microbial guilds that influence carbon and nitrogen cycling need to be accounted for to better predict $CH_4$ fluxes at landscape scales. Higher methane production occurring near the freshwater boundary with slight salinization (and sulfate incursion) might result from increased sulfate-reducing fermenter and syntrophic populations, which can produce substrates used by methanogens. Moreover, higher salinities can solubilize ionically bound ammonium abundant in the lower salinity wetland soils examined here, which could inhibit methanotrophs and potentially contribute to greater $CH_4$ fluxes observed in oligohaline sediments.

**IMPORTANCE**  Low-level salinity intrusion could increase CH4 flux in tidal freshwater wetlands, while higher levels of salinization might instead decrease CH4 fluxes. High CH4 emissions in oligohaline sites are concerning because seawater intrusion will cause tidal freshwater wetlands to become oligohaline. Methanogenesis genes alone did not account for landscape patterns of CH4 fluxes, suggesting mechanisms altering methanogenesis, methanotrophy, nitrogen cycling, and ammonium release, and increasing decomposition and syntrophic bacterial populations could contribute to increases in net CH4 flux at oligohaline salinities. Improved understanding of these influences on net CH4 emissions could improve restoration efforts and accounting of carbon sequestration in estuarine wetlands. More pristine reference sites may have older and more abundant organic matter with higher carbon:nitrogen compared to wetlands impacted by agricultural activity and may present different interactions between salinity and CH4. This distinction might be critical for modeling efforts to scale up biogeochemical process interactions in estuarine wetlands.

**KEYWORDS**  methane, methanogenesis, methanotrophs, salinity, sulfate, carbon cycling, decomposition, wetlands

T he carbon sequestration potential of vegetated estuarine ecosystems, referred to as blue carbon, has been the subject of considerable interest as a climate mitigation strategy (1–3). Tidal salt and brackish marshes are the most prevalent of the blue

Address correspondence to Susannah G. Tringe, sgtringe@lbl.gov.

The authors declare no conflict of interest.

See the funding table on p. 17.

carbon habitats (salt marshes, mangroves, and seagrass meadows) in the United States and therefore represent key targets for both preservation and restoration. Rates of carbon (C) burial per unit area of tidal wetlands greatly exceed those in all upland terrestrial ecosystems (4); indeed, despite their relatively small area, the total C sequestered annually in tidal wetlands has been estimated to be as high as that in tropical rainforests. Yet the carbon storage potential of estuarine wetlands may be threatened by rising sea levels and consequent inundation and salinization (5, 6). Salinization of estuarine wetlands may result from sea-level rise-driven intrusion of seawater or from decreased freshwater flows during droughts, which may further interact with urbanization and agricultural nutrient loading in many major estuaries (7–12). Salinization may imperil net soil carbon storage in estuarine wetland habitats by both reducing plant primary productivity (i.e., decreasing carbon inputs) and accelerating decomposition of soil carbon stocks (i.e., increasing carbon losses) (10, 13–16).

The impacts of salinity and salinization on production of the potent greenhouse gas methane ($CH_4$) are less well understood, particularly in tidal freshwater and brackish marshes. Salinity intrusion into freshwaters is hypothesized to suppress methanogenesis (17–19), as the additional sulfate in seawater may promote growth of sulfate-reducing bacteria, which are expected to outcompete hydrogenotrophic and acetoclastic methanogens for hydrogen and acetate based on the thermodynamic favorability of their respiratory pathways (13, 18, 20). However, observational studies and experimental tests in the field and laboratory of salinity effects on $CH_4$ in estuarine wetlands have yielded inconsistent results (21). Observational studies have generally, but not always, found a decrease in $CH_4$ flux with increasing salinity (18, 22). Field experiments in estuarine wetlands have found that soil C mineralization and $CH_4$ flux were suppressed by salinity intrusion (14) or not affected (23). Laboratory experiments with sediment cores from wetlands have shown decreases in $CO_2$ and $CH_4$ flux (24), increases in $CO_2$ flux concomitant with repression of $CH_4$ flux (25), or even increases in both $CO_2$ and $CH_4$ flux with salinity intrusion (16, 21). These mixed results suggest that more complex microbial responses and interactions control the effects of salinization on $CH_4$ fluxes.

Several studies to date have investigated effects of salinization on methanogen populations (26, 27), microbial communities (28), or microbial control of nutrient cycling (29, 30). However, these studies have largely focused on these dimensions individually, while microbial controls over $CH_4$ production are influenced by interactions with additional processes including decomposition, fermentation (31), methanotrophy (32), and nitrogen cycling (33, 34). Moreover, efforts to synthesize effects of estuarine salinity gradients and salinization on greenhouse gas fluxes suggest a need to better account for changes in electron acceptors, decomposition rates, alternative methanogenesis pathways (e.g., methyl dismutation and methyl reduction), and nutrient availability, as well as complex interactions among them (10, 13).

We sought to unravel ecosystem-scale relationships between salinity and $CH_4$ as a function of underlying microbial processes in wetlands spanning the natural salinity gradient across the San Francisco Bay estuary, including freshwater soils in the Sacramento-San Joaquin Delta (the Delta). Using both 16S rRNA gene data and shotgun metagenomic data, we compared microbial community features and $CH_4$ fluxes across the salinity gradient, including both reference and restored wetlands, which were paired where possible. We obtained soil $CO_2$ and $CH_4$ flux data from intact wetland soil cores, which were also used to characterize variation in soil chemistry and microbial community structure and function. Our objectives were to (i) determine patterns in $CH_4$ fluxes across the salinity gradient, including the influences of salinity and restoration; (ii) identify microbial metabolic pathways and taxa associated with $CH_4$ production; and (iii) assess interactions among methanogenic and non-methanogenic microbial functional groups (guilds) which might contribute to net $CH_4$ fluxes, along with the environmental drivers of those interactions. Understanding how microbial communities and biogeochemical processes change across this detailed salinity gradient will help us

predict how salinization will affect these parameters in the near and long terms, with implications for carbon storage and greenhouse gas emissions.

## RESULTS

### Soil CH$_4$ fluxes and biogeochemistry along the salinity gradient

Soil methane (CH$_4$) fluxes increased from freshwater to oligohaline (ca. 2.5-ppt salinity) wetlands but then decreased markedly across the increasing salinity of the San Francisco Bay and Delta following a broadly log-linear relationship (Fig. 1a and b; Fig. S1), with breakpoint regression showing a breakpoint at 1.4 ppt. This trend was driven by the markedly elevated emissions from the oligohaline Mayberry Farms restored wetland complex in the Delta, which were significantly greater than in most other locations. CH$_4$ flux varied significantly by individual site location [linear mixed effect (LME), $P < 0.05$] but not by wetland status (reference versus restored) (LME, $P > 0.05$). Restored wetlands did, however, have significantly higher CH$_4$ emissions in the Delta (LME, $P < 0.05$). CH$_4$ flux and soil respiration (CO$_2$ flux) were generally only loosely (but significantly) associated across the whole data set ($R^2 = 0.25$), while at Mayberry only they were more closely coupled ($R^2 = 0.88$, Fig. 1c). Soil chemistry and physical properties varied significantly along the salinity gradient, including percent soil C, which was highest in freshwater wetlands and broadly decreased with salinity, but also varied within and between wetland complexes (Fig. 1d). In turn, soil C was closely coupled to the relative abundances of nitrogen (N) and phosphorus (P) as expressed by N:P ratios, with increasing N:P in higher C (lower salinity) soils (Fig. 1e). High-C, low-salinity soils also had greater soil

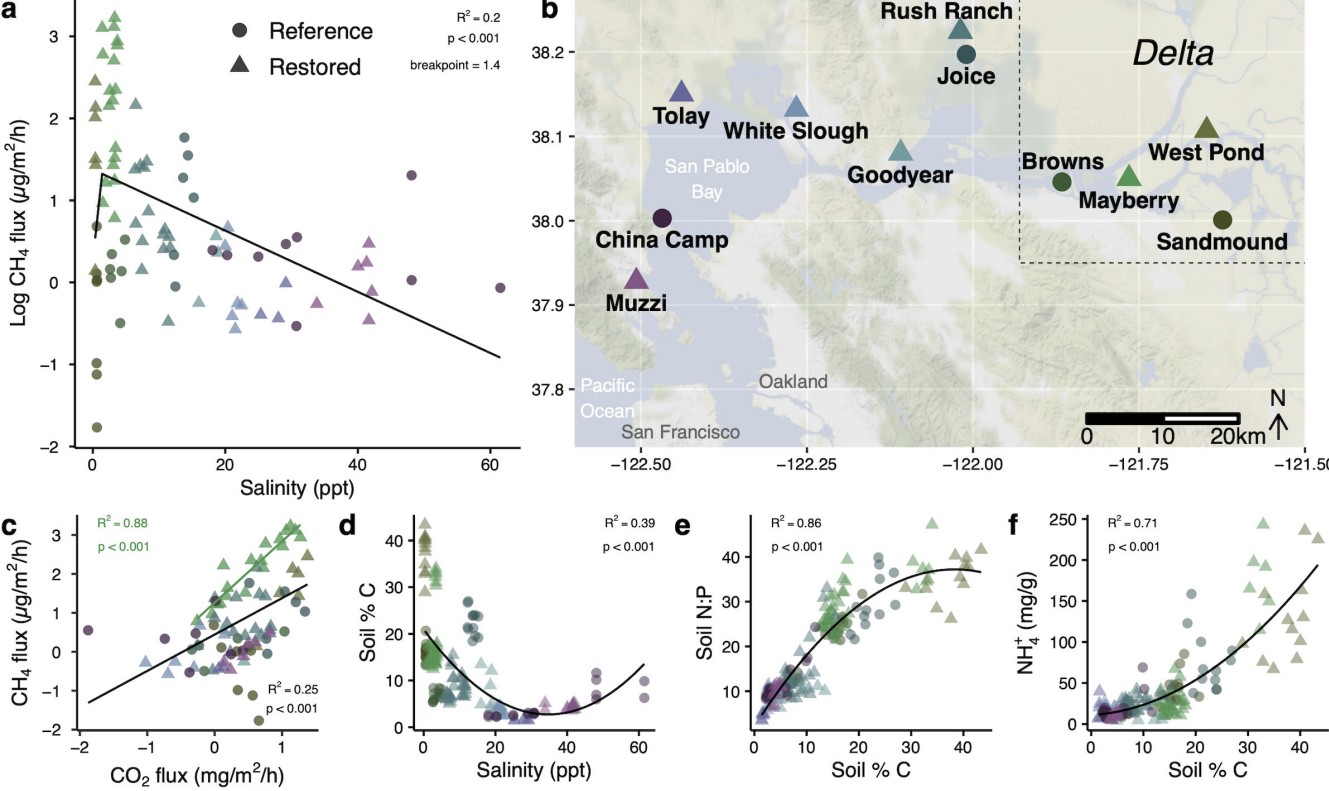

**FIG 1** Biogeochemical patterns in greenhouse gas fluxes and soil chemistry in wetland sites along the San Francisco Bay-Delta salinity gradient. (a) Variation in soil methane fluxes with salinity among wetland soils sampled along the gradient, with a segmented (breakpoint = 1.4 ppt) linear regression line (b) spanning the northern San Francisco Bay and Delta. (c) Relationship between soil methane and carbon dioxide fluxes, with linear regression lines shown separately for the Mayberry Farms wetland (green) and for all sites (black). (d) Soil carbon pools varied with salinity, and in turn, (e) soil N:P and (f) soil ammonium varied as a function of soil C in relation to salinity. Points are colored by site as in panel b; circles are reference wetlands and triangles are restored wetlands. Regression lines in panels d–f are from second-order polynomial regressions, which fit the data better than linear regressions.

ammonium pools (Fig. 1f). Soil C, N:P, and ammonium were not significantly affected by depth (LME, $P > 0.05$).

Across all the sites, $CH_4$ fluxes were most positively correlated with soil $CO_2$, soil C and N:P, and dissolved organic carbon (DOC) (Table S4) and were negatively correlated with water salinity as well as soil sulfate ($SO_4^{2-}$) and chloride ($Cl^-$). However, when considering only Delta sites, $CH_4$ was not correlated with $SO_4$ and only weakly (positively) correlated with salinity and chloride concentrations (Table S5). $CH_4$ fluxes in the Delta were, as across all sites, positively correlated with $CO_2$, soil N:P, and DOC, and inversely correlated with soil volumetric water-filled pore space, a measure of bulk density and moisture content (Table S5). Among all sites, the strongest combined predictors of $CH_4$ fluxes based on least absolute shrinkage and selection operator (LASSO) variable importance scores were DOC, $CO_2$ flux, and soil N:P ratios (Fig. S2a). However, DOC was a poor predictor of $CH_4$ fluxes in a similar LASSO model using only Delta sites, which had the highest CH4 emissions and where $CO_2$, total nitrogen, salinity, and $SO_4^{2-}$ were the strongest *positive* predictors of $CH_4$ flux (Fig. S2b), implying an increasing influence of seawater was associated with *higher* $CH_4$ fluxes in the Delta. Volumetric water-filled pore space and soil $NO_3:NH_4$ ratios were negative predictors of $CH_4$ fluxes in models for the Delta and among all sites (Fig. S2).

## Microbial metabolic genes and wetland salinity

With increasing salinity, we observed general increases in sulfur cycling gene relative abundances and decreases in methanogenesis gene relative abundances (Fig. 2b; Fig. S3). Yet among sulfur cycling genes, only *satA* (catalyzing the first step in both assimilatory and dissimilatory sulfur (S) reduction) was strongly correlated with salinity (Fig. S3). Sulfate reduction genes *dsrAB* (for dissimilatory sulfite reductase, converting sulfite to sulfide) were weakly correlated with salinity and moderately correlated with $SO_4$ across all of our sites (Fig. 2b; Table S6). However, sulfate reduction genes *aprAB* (APS reductase, downstream of satA) were moderately correlated with salinity and, like *satA*, were strongly correlated with $SO_4$ (Tables S6 and 7). Although several genes for $CH_4$ metabolism (*mcrABG* and *mtrCDEFG*) were negatively correlated with salinity, these relationships were not especially strong, particularly within Delta wetlands (Fig. 2b; Fig. S3: Tables S6 and S7). Most correlations between nutrient cycling genes and salinity were even stronger among reference wetlands than across all sites (Fig. S3); this was also true for genes for aromatic utilization, denitrification, acetoclastic methanogenesis, and $CH_4$ oxidation. A notable exception to this trend was the gene *mttB* (methanogenic reduction of trimethylamines), which increased with salinity across all soils (Fig. S3).

Several nitrogen cycling genes also varied with salinity and were more abundant in freshwater reference wetlands than nearby restored sites (Fig. S3), including assimilatory and dissimilatory nitrate reductases (*narBH*, *nasB*, and *napA*), and ammonia oxidation and assimilation genes (*pmoBC*, *hao*, and *aspQ*). However, some nitrate reductases (*narBH* and *nasB*) were, like $CH_4$ flux, highest in oligohaline sites (Fig. S3), as was the gene for utilization of the compatible solute trehalose (*treA*, Fig. S3).

## Metabolic genes linked with $CH_4$ fluxes

Most element cycling genes were more correlated with $CH_4$ among the subset of Delta sites than in all sites or only reference wetlands (Fig. 2; Fig. S4), suggesting different forces may govern $CH_4$ fluxes in lower salinity wetlands where $CH_4$ fluxes were highest. Although central methanogenesis (*mcrABG* and *mtrCDEFG*) and hydrogenotrophic (*fwdDF* and *mtd*) genes were strongly positively correlated with $CH_4$ fluxes across all sites, the highest flux sites did not have the greatest relative abundances of these genes (Fig. 2; Fig. S4). Genes for the consumption of $CH_4$ (*pmo*C) were negatively correlated with $CH_4$ fluxes (Fig. 2; Fig. S4), although *pmoABC* genes cannot be differentiated from closely related ammonia oxidation genes (*amoABC*) in the Metagenomics Rapid Annotation using Subsystems Technology (MG-RAST) annotations. Using the Tree-based Sensitive and Accurate Phylogenetic Profiler (TreeSAPP) to assign *pmoA-amoA* genes to

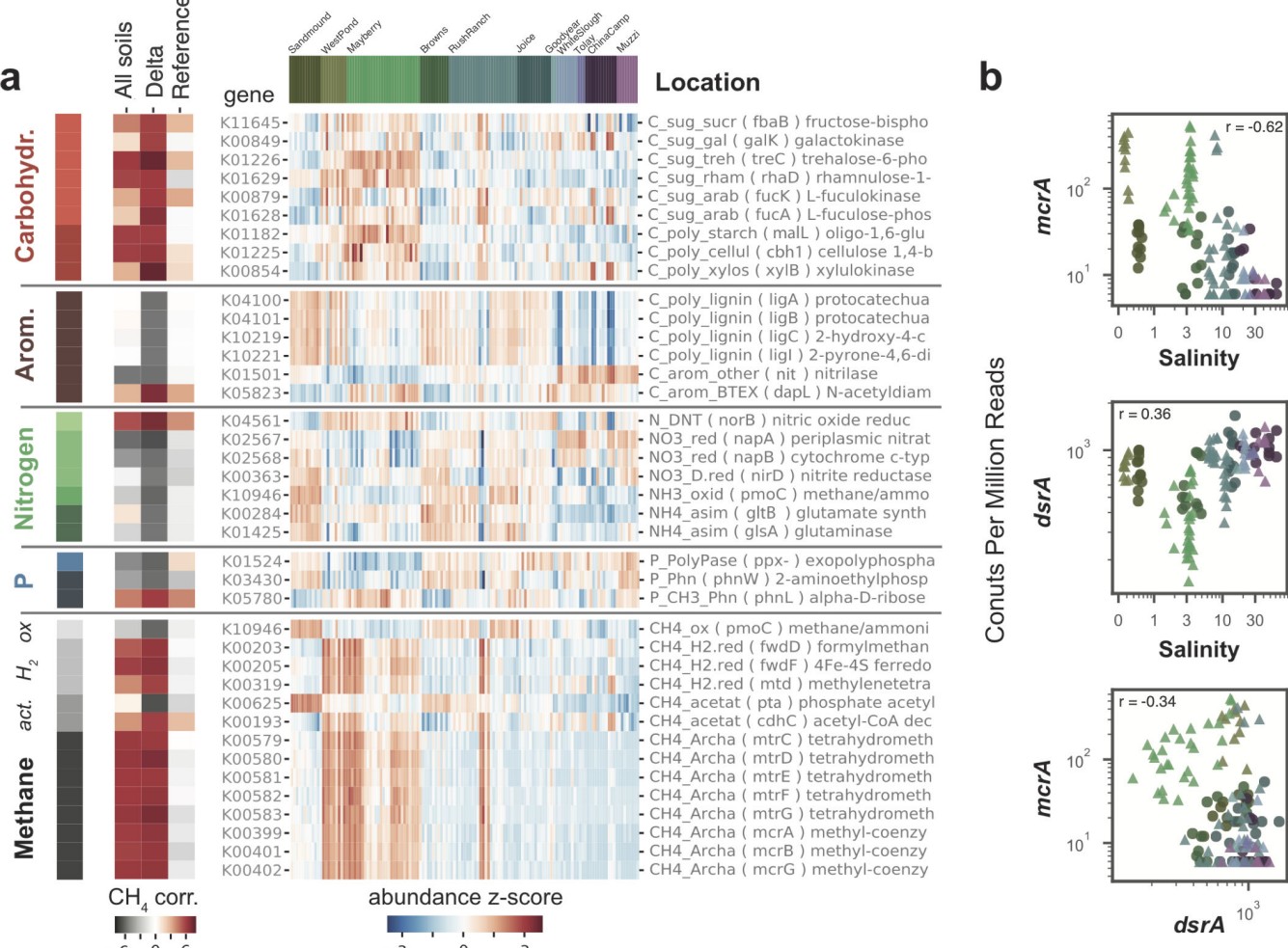

**FIG 2** Functional gene relative abundances significantly correlated with CH₄ flux. (a) Relative abundances of element cycling genes, color coded by element cycle and pathway (carbohydrates, aromatics, nitrogen, phosphorus, methane—acetoclastic, hydrogenotrophic, and oxidation), across sites, with site (ordered from low to high salinity) indicated by the bar at the top of the heatmap, which matches colors in the map of locations in Fig. 1b. Correlations with methane fluxes are shown in the leftmost heatmap based on all sites, or subsets of sites corresponding to only sites in the Delta, or to only reference wetland sites. (b) Scatterplots of relative abundances of selected genes for methanogenesis and sulfate reduction vs salinity and vs each other.

methane-oxidizing bacteria, ammonia-oxidizing archaea (AOA), or ammonia-oxidizing bacteria (AOB) and therefore to differentiate *pmoA* and *amoA*, we found that Class IIa MOB *pmoA* relative abundances were negatively associated with CH₄ emissions in the Delta, and AOB *amoA* relative abundances were negatively associated with CH₄ both across the whole data set and within the Delta (Fig. 3). Furthermore, total *pmoA* and total *amoA* relative abundances were negatively associated with each other (Fig. 3). Sulfate reduction genes *dsrAB* were moderately anti-correlated with CH₄ fluxes but poorly correlated with methanogenesis genes *mcrAB*, especially in Delta soils where their relationship appeared positive in restored sites (Fig. 2b; Fig. S5b; Tables S6 and S7). Although *aprAB* genes for an early step in sulfate reduction showed stronger negative relationships with *mcrAB* across all sites than *dsrAB*, these relationships were still weak in Delta soils (Tables S6 and S7). Contrastingly, some genes for S assimilation (*cysC* and *sir*) had stronger negative relationships with methanogenesis genes both among all sites and in the Delta, and these genes were negatively correlated with suflate reduction genes (Tables S6 and 7), while *sat* genes for the first reaction in sulfate reduction were inversely related to methanogenesis pathways across all sites and to a lesser extent in soils of the Delta (Tables S6 and 7).

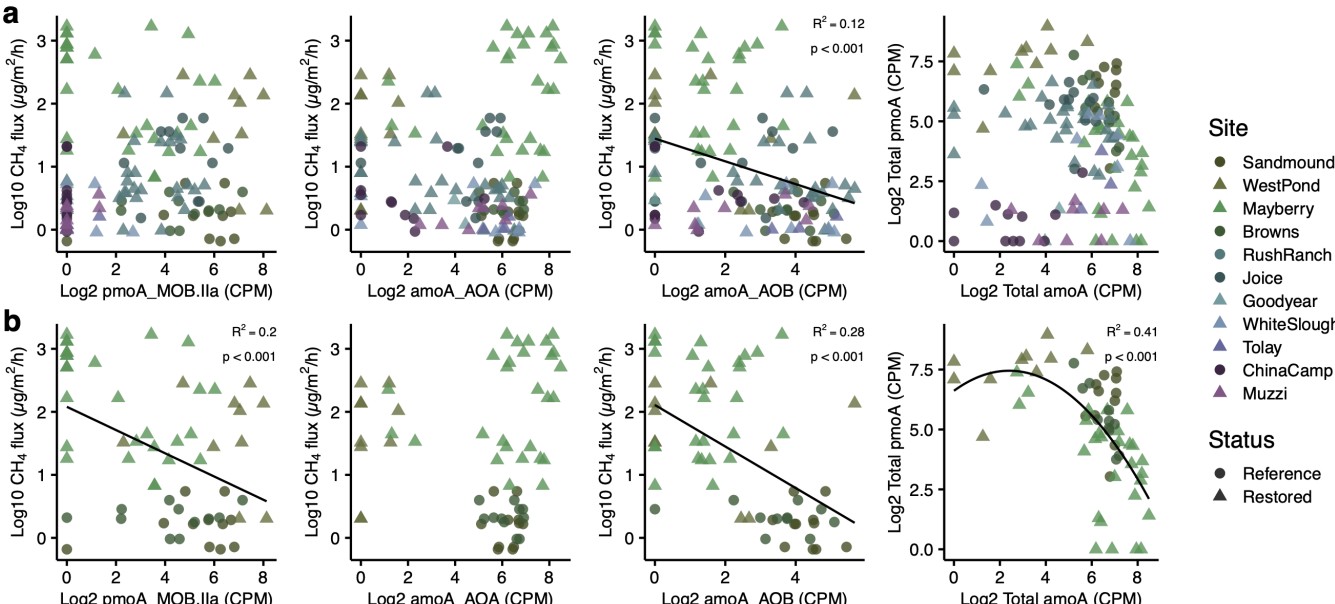

**FIG 3** Relationships between CH$_4$ flux and relative abundance of *pmoA-amoA* genes assigned to Class IIa methane-oxidizing bacteria (pmoA_MOB.IIa), ammonia-oxidizing archaea (amoA_AOA), or ammonia-oxidizing bacteria (amoA_AOB) across the whole data set (a) and in the Delta (b). Also shown are relationships between total *pmoA* relative abundance (*pmoA-amoA* assigned to MOB) and total *amoA* (*pmoA-amoA* assigned to AOA and AOB) relative abundance. Annotation and taxonomic assignment were done with TreeSAPP. Lines are shown when relationships are significant. CPM, counts per million.

Microbial carbon cycling genes revealed marked shifts from utilization of aromatic compounds to carbohydrates as CH$_4$ fluxes increased, particularly in Delta soils (Fig. 2a; Fig. S4). Lower CH$_4$ fluxes were associated with genes for lignin degradation (*ligABCL*), while higher fluxes were associated with breakdown of hemicellulose (*xylB*), cellulose and starches (*cbh1* and *malL*), and sugars including sucrose, galactose, rhamnose, and arabinose (*fbaB*, *galK*, *rhaD*, and *fucAK*, respectively). Trehalose degradation (*treC*) was also associated with CH$_4$ fluxes in the Delta, although this sugar is also a compatible solute which may reflect osmotic adaptation. Several N cycling genes were negatively correlated with CH$_4$ in the Delta, including nitrate reductases (*napAB* and *nirD*), and ammonia oxidation (*amoC/pmoC*) and assimilation genes (*glsA* and *gltB*), while nitric oxide reductase (*norB*) was positively correlated with CH$_4$ (Fig. 2a).

## Microbial communities and methane

Across all the wetland soils studied, microbial community composition determined by the 16S rRNA gene (Fig. 4b) was structured by restoration status, salinity, and nutrients (Fig. 4c). Wetland site accounted for much of the variation in microbial community composition [Fig. 4b; permutational multivariate analysis of variance (PERMANOVA), $R^2 = 0.66$, $P = 0.001$], while vegetation type as a second predictor variable had a lesser effect (Fig. S6; PERMANOVA, $R^2 = 0.06$, $P = 0.001$). Several soil features were also closely linked with microbial community composition, including salinity, bulk density, C, and N, among others (Fig. 4c). Bacterial communities were dominated by several classes of Proteobacteria and the phyla Acidobacteriota, Actinobacteriota, Bacteroidota, Chloroflexi, Firmicutes, and Nitrospirota (Fig. 4b). Archaeal phyla were less abundant, but included Halobacteriota, Crenarchaeota, Euryarchaeota, Aenigmarchaeota, Altiarchaeota, Asgardarchaeota, Nanoarchaeota, and Thermoplasmatota (Table S8).

Soil CH$_4$ fluxes were positively correlated with the phyla Firmicutes (including several Bacilli and Clostridia taxa) and Spirochaetota (Fig. 4a; Table S8), along with Halobacteriota (including Halobacteriales and Methanosarcinales) and Chloroflexi (particularly class Dehalococcoidia). Within the less correlated Actinobacteriota phylum, the orders Frankiales, Micrococcales, and Pseudonocardiales had several members well correlated

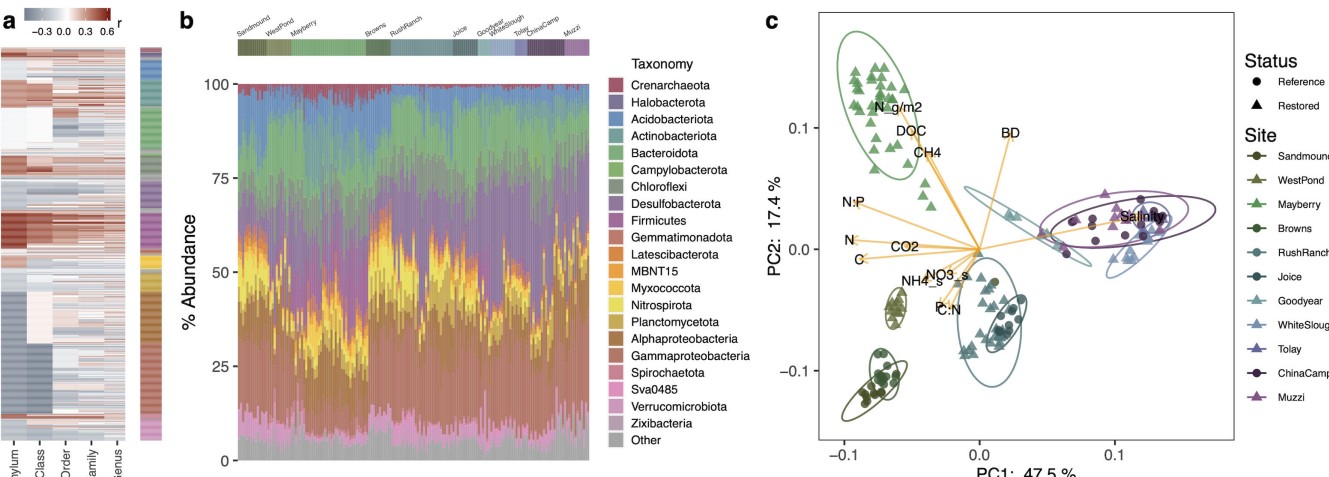

**FIG 4** Correlations of microbial taxa with soil methane fluxes (a), microbial community composition based on 16S rRNA gene sequencing (b), and principal component analysis (PCA) of Aitchison distance (c). The heatmap in panel a shows the Pearson correlation coefficient of microbial taxa with soil methane fluxes calculated at each taxonomic rank, with taxonomic groups indicated by the colored bar at right with colors corresponding to the legend in panel b. Details of those results are presented in Table S8. Relative abundances (proportion of total sequence reads after normalization by DESeq2) of major microbial groups (b) are shown for the most abundant phyla (along with *Proteobacterial* classes) for each sample, with study site indicated by the bar at the top, colored to match location data in panel c. PCA ordination in panel c shows clustering of soil microbial communities by wetland location (PERMANOVA, $R^2$ = 0.66), along with projected loadings of environmental chemistry data into the reduced dimensional space. C, N, and P are total elements in soil. Salinity was measured *in situ* in soil coring holes. Ellipses show 95% confidence intervals around the centroid. Abbreviations: BD, bulk density; DOC, dissolved organic carbon.

with $CH_4$ fluxes. The phyla Verrucomicrobiota and Desulfobacterota were negatively correlated with $CH_4$ fluxes, as were the phylum Proteobacteria and the order Gammaproteobacteria, though several proteobacterial families (Geminicoccaceae) and genera (*Sphingomonas* and *Ellin6055*) were instead positively correlated with $CH_4$ fluxes (Fig. 4a; Table S8).

## Functional guilds and methane

Functional guilds of microbes involved in carbon, nitrogen, and sulfur cycling were obtained from the taxonomic composition of metagenome sequence reads annotated as encoding the function, or from 16S rRNA gene taxonomy in cases of known vertical inheritance and close relationships between phylogeny and biochemical function. For example, the majority of cellulose degrading *cbh-1* genes were assigned to Firmicutes or Actinobacteriota, explaining the role of these $CH_4$-correlated taxa in contributing to this $CH_4$-linked degradation pathway (Fig. S7). However, due to the higher statistical resolution and broader site coverage of 16S rRNA data, as well as ambiguities and/or low numeric counts of some genes [beneath quantitative thresholds (35)] in metagenome data, 16S rRNA gene relative abundances were deemed more informative for determining quantitative relationships. Functional guilds from 16S rRNA gene taxonomy were generally consistent with their relative abundance in shotgun sequence-derived annotations using TreeSAPP, although TreeSAPP did not appear to reliably identify *amo* genes from nitrite oxidizers or *pmo* genes from Type IIb $CH_4$ oxidizers (Fig. S8).

Unlike the patterns across all sites, $CH_4$ fluxes in the Delta were not strongly positively correlated with any methanogenic genera or genera from any other functional guild. On the other hand, there were several genera from various functional guilds strongly negatively correlated with $CH_4$. These include the ammonia-oxidizing archaeal genera *Nitrosarchaeum* and *Cand. Nitrosotenuis*, several genera of ammonia-oxidizing bacteria, and several nitrite-oxidizing bacteria (NOB) (*Nitrospira*, *4–29-1* identified only to class level, and *P9 × 2b3D02* identified only to class level) (Fig. S9). Some sulfur-reducing (*Desulforhabdus*) and sulfur-oxidizing (Thioalkalispiraceae identified to family level) taxa were strongly negatively correlated with $CH_4$ flux (Fig. S9).

LASSO regression modeling of $CH_4$ fluxes based on these guild members was used to assess their relative importance in contributing to $CH_4$ fluxes (LASSO regression model, $R^2 = 0.81$). Guild members most positively associated with $CH_4$ fluxes in the Delta included the iron oxidizer *Leptolinea*, the methane oxidizer *Methyloceanibacter*, and the sulfur oxidizer *Thiobacillus*, while the most negatively associated genera included the sulfate reducers *Desulfobacca*, *Sva1033* (identified to family level), *Desulfomonile*, and *Desulfatiglans*, the methylotroph *Methylotenera*, and the methane oxidizer *Methylocystis* (Fig. S10a). Similarly, when relative abundances of these guilds were considered in aggregate (LASSO regression model, $R^2 = 0.65$), the sulfate reducers and Type I and IIa methanotrophs were the most negatively correlated with $CH_4$, while iron oxidizers, acetoclastic methanogens, and sulfur-oxidizing bacteria were most positively correlated with $CH_4$ (Fig. S10b).

## Potential interactions among microbial guilds

To assess the influence of multiple microbial guilds correlated with $CH_4$ on net $CH_4$ fluxes, we compared the relative abundances of guilds with one another (and known mechanisms) to assess their potential interactions. Strikingly, the relative abundance of methanogens was significantly higher at the freshwater West Pond restored wetland than at the nearby oligohaline Mayberry Farms site (Fig. 5), despite $CH_4$ fluxes that were up to an order of magnitude higher at Mayberry Farms (Fig. 1a). This was also true of the relative abundance of central methanogenesis genes in shotgun metagenomic data from these sites (Fig. 2a; Fig. S4). However, the relative abundance of methanotrophic bacteria was also significantly higher in West Pond soils, while populations of AOA were significantly higher in Mayberry Farms. While AOA were also present in Sandmound and Brown's Island, reference freshwater and oligohaline Delta wetlands, respectively, these soils also had significantly more abundant AOB and NOB compared to the adjacent restored wetlands (Fig. 5). Methanotrophic genera as a whole declined in relative abundance across the salinity gradient and also shifted in composition, with certain genera more abundant in the Delta (e.g., *Methylocystis* and *Crenothrix*) and others more abundant at the more saline sites (e.g., *Methyloceanibacter*) (Fig. S11).

To evaluate potential effects of guild interactions on net $CH_4$ fluxes, we constructed a series of structural equation models (SEMs), based on aggregate guild relative abundances derived from 16S rRNA gene taxonomy. The most common mechanistic predictors of $CH_4$ fluxes in LASSO models were combined acetoclastic and mixotrophic methanogens (CH4 ac+mix) and Type IIa methanotrophs (MOB IIa), along with $CO_2$ flux (an indicator of decomposition) and soil bulk density. We then used composite models to incorporate simultaneous predictions of acetoclastic and mixotrophic methanogens and Type IIa methanotrophs into base SEMs. The model with these SEM "branches" was not significant across all sites, but was significant ($P > 0.05$) in the Delta sites (Fig. 6). The final SEM model for the Delta shows acetoclastic and mixotrophic methanogens and $CO_2$ flux predicting the composite variable "methane generation"; water-filled pore space, AOA, and NOB predicting MOB IIa; MOB_IIa and bulk density predicting the composite variable "methane oxidation"; and then methane generation and methane oxidation predicting the observed methane flux (Fig. 6).

## DISCUSSION

### Estuary-scale patterns in wetland methane fluxes

Our first objective was to understand the patterns and drivers of $CH_4$ fluxes across an estuarine salinity gradient. $CH_4$ fluxes exhibited a broadly log-linear relationship with soil salinity across the estuarine gradient (Fig. 1a). However, the highest $CH_4$ fluxes occurred in oligohaline wetlands, in agreement with ecosystem-scale eddy covariance observations at the same restored Delta wetland sites (22, 36–39) and a previous meta-analysis of tidal marsh soils (18) (Fig. S1). Although the highest $CH_4$-producing soils in our study were non-tidal restored wetlands, this concordance suggests that maximum

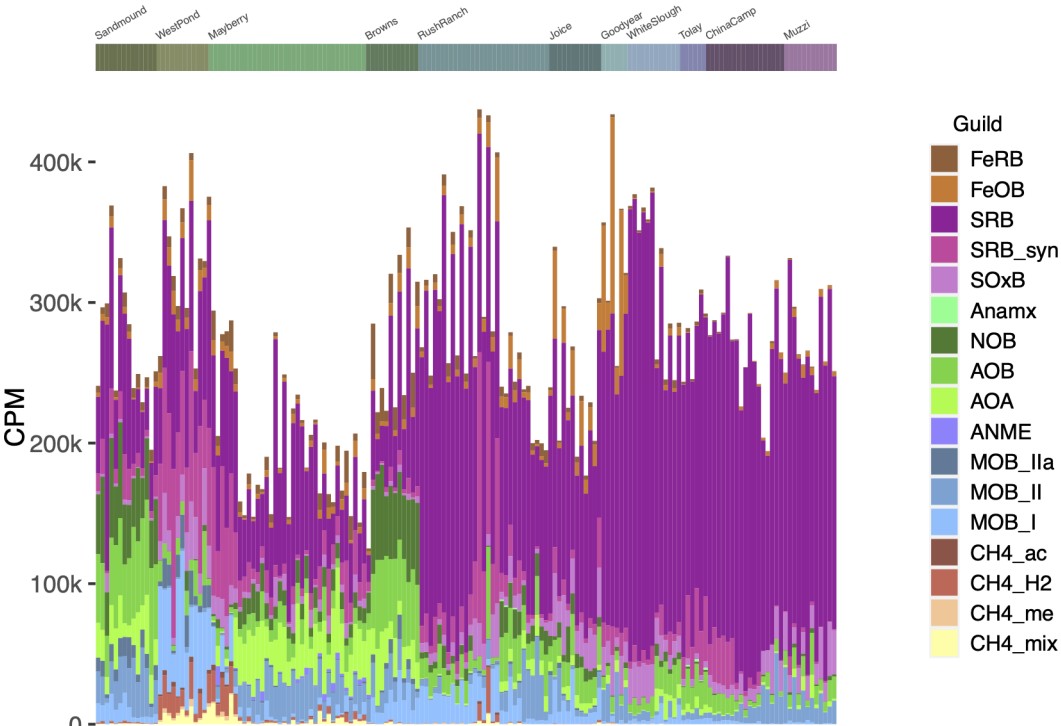

**FIG 5** Relative abundance of microbial guilds based on 16S rRNA gene taxonomy. Guild relative abundances are shown as counts per million 16S rRNA gene sequence reads following normalization using DESeq2's variance stabilizing transformation to account for differences in read depth among samples. Guilds were assigned based on taxonomy in published review papers and were iron-reducing and iron-oxidizing bacteria (FeRB and FeOB, respectively); sulfate-reducing bacteria, syntrophs, and sulfur-oxidizing bacteria (SRB, SRB_syn, and SOxB); anaerobic ammonia-oxidizing bacteria (Anamx); nitrite and ammonia-oxidizing bacteria and ammonia-oxidizing archaea (NOB, AOB, and AOA); anaerobic methane-oxidizing archaea (ANME); Type I, II, and IIa methanotrophic bacteria (MOB_I, MOB_II, and MOB_IIa); and acetoclastic, hydrogenotrophic, methyl-reducing, and mixotrophic methanogens (CH4_ac, CH4_H2, CH4_me, and CH4_mix). CPM, counts per million.

net $CH_4$ fluxes may occur at low but non-zero salinities (40–42), congruent with studies demonstrating an increase in $CH_4$ following low-level salinization (ca. 5 ppt) of some freshwater wetland soils (16, 21, 43).

These patterns of $CH_4$ fluxes did not strongly support the hypothesis that competition for carbon substrates from sulfate reducers with increasing seawater influence (increased salinity and sulfate) is the primary factor influencing archaeal methanogenesis and $CH_4$ fluxes in estuarine wetlands (14, 18, 25, 43). Although $CH_4$ fluxes were negatively associated with sulfate and salinity across the full salinity range studied (Table S4), in higher $CH_4$ Delta sites, $CH_4$ was not correlated with $SO_4$ and was weakly positively associated with salinity (Table S5). A case in point is the Mayberry Farms location; in the same wetland complex with relatively little spatial variation in salinity (1.4–3.7 ppt), $CH_4$ emissions varied by over three orders of magnitude (6–1,680 μg/m²/h), indicating that other variables besides sulfate reducer activity exert a strong influence on net $CH_4$ emissions. $CH_4$ flux was positively associated with DOC, soil N:P ratios, and $CO_2$ fluxes across all sites and within the Delta (Tables S4 and 5). Mayberry Farms, in particular, showed a remarkably strong relationship between $CH_4$ and $CO_2$ fluxes (Fig. 1c), which could suggest a dominant influence of overall organic carbon decomposition rates on $CH_4$ production, and/or a lack of $CH_4$ oxidation at this site, or a greater contribution of methylotrophic or acetoclastic methanogenesis, both of which produce $CO_2$ (44). 16S and metagenomic data, however, suggest similar correlations between different methanogen guilds and $CH_4$ emissions in the Delta and the whole data set (Fig. S5). As salinity increases, non-competitive methyl-based substrates such as trimethylamine, which can function as compatible solutes or are degradation products

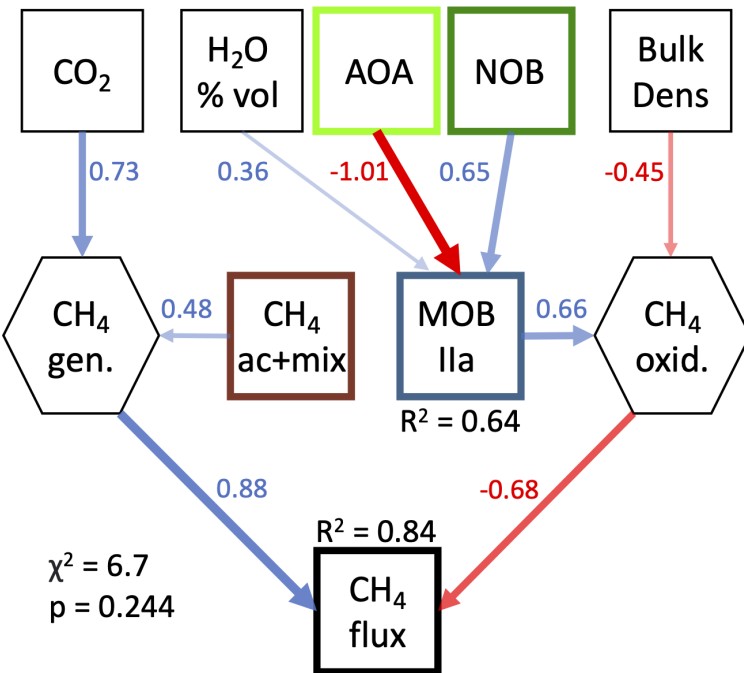

**FIG 6** Structural equation model predicting soil methane fluxes as a function of microbial guilds and soil $CO_2$, using data from sites from the Delta only. Guild abbreviations and colors match those in Fig. 5, except that here, acetoclastic and mixotrophic methanogens were combined (CH4 ac+mix). Regression fit coefficients ($R^2$) indicate the degree of prediction for each predicted feature in the model, and hexagons indicate composite variables. Arrows indicate directional relationships based on known interactions, with blue arrows indicating positive relationships and red arrows indicating negative relationships. Arrow widths are proportionate to the strength of these relationships, also shown by embedded numbers which are scaled model coefficients. Note that here, the $P$ value above 0.05 indicates a significant model.

of compatible solutes (45, 46), may increase and contribute a greater proportion of the $CH_4$ than in freshwater environments. While the increase in *mttB* relative abundance with salinity partially supports this hypothesis, methyl-reducing and mixotrophic taxa relative abundances did not increase with salinity (Fig. S5). High $CH_4$ emissions at that site during our sampling may also be attributed to its recent conversion to a restored wetland and the subsequent increase in primary productivity and labile carbon inputs; $CH_4$ emissions have since declined both there and at West Pond over time (2012–2020) (47). High $CH_4$ production has also been observed in oligohaline wetlands in the Delaware River estuary where isotopic measurements indicated patterns across the salinity gradient were driven more by a lack of $CH_4$ consumption than by greater gross $CH_4$ production (42).

The supply of methanogenic substrates can depend on the overall rate of decomposition in soils, which may in turn be shaped by variation in nutrient availability (13, 48, 49). In our wetland soils, total C pools broadly declined with salinity (Fig. 1d), and lower percent C was associated with lower N:P ratio and $NH_4^+$ concentrations (Fig. 1e and f; Table S4). These patterns likely arise from lower primary productivity and greater rates of decomposition in more saline wetlands (15, 48, 50), combined with tightly constrained soil C, N, and P stoichiometry (51, 52). Higher soil N:P ratios and inorganic N availability in our freshwater and oligohaline soils (Fig. 1e and f) may further arise from mechanisms promoting increasing N (vs P) limitation with greater salinity in estuaries, including sulfate-driven loss of P sorption (leading to higher P availability) and decreased N availability due to inhibition of N fixation (53–56). Notably, the larger pools of extractable ammonium in our freshwater wetland soils (Fig. 1f) might be susceptible to desorption by ionic exchange with salinity intrusion (16, 42, 57–59), an effect which may depend in part on land use history and agricultural runoff (7).

## Microbial taxa, metabolism, and methane fluxes

Within Delta soils, many methanogenic taxa and genes for $CH_4$ production were more abundant in the lower $CH_4$-emitting freshwater restored wetland than in the highest $CH_4$-emitting oligohaline sites (Fig. 2b; Fig. 4), again suggesting other factors contribute to soil $CH_4$ fluxes. It is important to note, however, that we did not measure gene or protein expression, which could potentially show different patterns than metagenome-based relative abundances; future work using metatranscriptomics and/or metaproteomics would provide valuable additional information. Utilizing both 16S rRNA gene taxonomic data and shotgun metagenomic data, we consider how five factors—$CH_4$ oxidation, nitrogen cycling, carbon degradation, sulfur cycling, and iron cycling—may contribute to the observed net $CH_4$ fluxes.

Genes essential to methanotrophy (*pmo*) were negatively correlated with $CH_4$ fluxes, particularly in the Delta (Fig. 2; Fig. S4d). Yet *pmo* and ammonia monooxygenase (*amo*) gene functions are not differentiated in common gene ontologies or annotation pipelines (60–63) due to their close evolutionary relationship (64–67). Instead, we used phylogenetic placement and taxonomic assignment to distinguish microbes that oxidize $CH_4$ from those that oxidize ammonia (Fig. 3 and 5; Fig. S5). The relative abundance of *pmoA* genes assigned to Class IIa MOB was negatively correlated with $CH_4$ flux in the Delta (Fig. 3b; Fig. S5d), as was the relative abundance of *amoA* genes assigned to AOB (Fig. 3; Fig. S5c and d). In the 16S amplicon data, methane-oxidizing bacterial guilds as a whole were negatively correlated with $CH_4$ fluxes (Fig. S5f), and two $CH_4$ oxidizing genera were negatively correlated with $CH_4$ fluxes in the Delta (Fig. S9), supporting the potential for methanotrophs to substantially alter net wetland $CH_4$ fluxes (68–71). Incongruous patterns of methanogen guilds and $CH_4$ fluxes, particularly at our highest $CH_4$ sites (Fig. 5), further suggested the potential importance of methanotrophy in our wetland soils. For example, Mayberry had higher $CH_4$ emissions than West Pond (Fig. 1a) despite less abundant methanogens (Fig. 5); this may be in part due to more abundant methanotrophs at West Pond consuming much of the $CH_4$ and leading to less net $CH_4$ flux relative to Mayberry. Alternatively, soil temperatures at Mayberry in 2013 were generally 2–3°C higher than at West Pond, which could increase methanogenesis rates (72). Detailed soil temperature data are not available for other sites.

If methanotrophy exerts an important influence on net $CH_4$ flux, it is then also important to consider nitrogen cycling, as studies in agricultural fields, forests, and rice paddies have shown that $CH_4$ oxidation can be inhibited by excess ammonium and nitrite (33, 34, 73–76). Interactions between inorganic N availability and methanotrophs may depend on both the form of N and the community of microbes present (33). Some methanotrophs (particularly Type IIa MOB) may be sensitive to ammonia due to their inability to detoxify hydroxylamine, the immediate metabolic product of ammonia oxidation (74, 77–81), and may also be inhibited by nitrite accumulation (80–83). Shifts in methanotrophic community dominance from Type II to Type I MOB have been observed with excess inorganic N, including ammonium and nitrite, in several environments (33, 73, 82, 84). The soluble (bioavailable) fraction of ammonium is particularly important for predicting the effects of ammonium on methane oxidation (85).

Indeed, across our sites, nitrogen cycling guild relative abundances generally opposed trends in methanotroph relative abundances, although correlations among those guilds were mixed (Fig. 5; Fig. S5e and f). MOB_I and MOB_IIa were negatively correlated with AOA, particularly in the Delta (Fig. S5e and f). Higher ratios of ammonia oxidizers (generating nitrite) to nitrite oxidizers (consuming nitrate), which may be inhibited at high ammonia concentrations (86), are expected to lead to the accumulation of soil nitrite ($NO_2^-$), which could inhibit methanotrophy. Greater ammonia oxidizer (AOA + AOB):NOB ratios were also weakly associated with greater methanogen:methanotroph ratios across all sites ($R^2 = 0.02$, $P = 0.07$) and in the Delta ($R^2 = 0.08$, $P = 0.01$) (Fig. S12). Increases in this ratio have been associated with higher $CH_4$ flux in previous studies (87, 88), but the weak relationship observed in our data suggests that this exerts only a minor influence on $CH_4$ flux across our sites.

More specifically, high-$CH_4$, low-methanotroph Mayberry soils had abundant populations of AOA, unlike the lower-$CH_4$, high-methanotroph West Pond soils. While the AOAs were also prevalent in reference freshwater and oligotrophic wetlands (Sandmound, Brown's Island), at those sites they were accompanied by even greater populations of NOB, suggesting that ammonia at those sites could be fully oxidized to nitrate in contrast to the neighboring restored wetlands (Mayberry, West Pond) in which NOB were nearly absent (Fig. 5). These results linking methanotrophs, ammonia oxidizers, and nitrite oxidizers are broadly consistent with descriptions of inhibition of $CH_4$ oxidation by excess N (33, 34), and we suggest that this effect might be linked with greater net $CH_4$ fluxes in our wetland soils, particularly in the Delta (Fig. 1a and f).

Genes reflecting carbon substrate availability were tightly connected to patterns of $CH_4$ fluxes, particularly in the Delta, as were cellulose degrading taxa. Higher $CH_4$ fluxes were positively correlated with several genes associated with degradation of plant biomass, including those connected to metabolism of simple sugars, cellulose, and hemicellulose, and inversely correlated with genes for lignin degradation (Fig. 2; Fig. S4a and b). $CH_4$ fluxes were correlated with most members of the phylum Firmicutes and several members of the Actinobacteriota (Fig. 4A) which were the dominant cellulose degraders in our wetlands (Fig. S7), a function we suggest is linked to measured $CH_4$ fluxes (Fig. 2). These patterns suggest $CH_4$ fluxes may be constrained by soil C availability or the overall rate of decomposition in soils (49, 68, 89, 90), both of which are affected by plant productivity and root exudation (68); this is concordant with our finding that $CH_4$ flux was highly correlated with $CO_2$ flux in our highest $CH_4$-producing site (Fig. 1). Although syntrophic bacteria lack a consistent genetic marker (91–94), we found that relative abundances of known syntrophic taxa were also linked to higher $CH_4$ fluxes (Fig. S5e and f; Fig. S9).

As was the case with the overall patterns in sulfate concentrations, the relative abundances of genes in sulfur cycling pathways showed limited support for the hypothesis that competitive inhibition of methanogens by sulfate-reducing bacteria (SRB) is a key driver of $CH_4$ fluxes. Overall, neither $CH_4$ flux nor methanogen relative abundance was strongly negatively correlated with sulfate concentrations or SRB relative abundance (Fig. 2b; Fig. S5a, c, and e), and weak correlations in the Delta sites were actually "positive" (Fig. S5b, d and f). Future work should include hydrogen sulfide ($H_2S$) measurements to examine the ratio of $H_2S$ to $CH_4$ as a terminal electron sink.

In addition to sulfate, oxidized iron ($Fe^{3+}$) represents another alternative electron acceptor for microbial metabolism in anaerobic sediments that is more energetically favorable than $CO_2$. Activity of iron reducers is expected to be negatively correlated with methane concentrations due to competition between iron reducers and methanogens (95). In our data set, the relative abundance of iron reducers was negatively correlated with $CH_4$ flux only in the Delta (Fig. S5), and this trend was driven primarily by taxa in the Geobacteraceae family (Fig. S10). These findings are in line with another study in the Delta that found higher iron concentrations in alluvium soils were correlated with lower ecosystem-scale $CH_4$ flux (96). In the context of salinization, previous work has found an initial increase in iron reduction rates and reduced iron ($Fe^{2+}$) concentrations following simulated seawater intrusion, but this effect diminished once the labile pool of iron oxides was consumed (97).

## Linking biogeochemistry, functional guilds, and $CH_4$ fluxes: SEMs

To integrate the interacting effects of multiple microbial guilds on soil $CH_4$ fluxes, we tested SEMs representing plausible metabolic interactions that determine net $CH_4$ fluxes (Fig. 6). Significant SEMs were found only for the Delta sites; SEMs for the whole data set were not significant, possibly due to non-linearities in the relationships between biogeochemical and microbial variables across such a broad salinity gradient. In the Delta, $CH_4$ fluxes were consistently predicted by the relative abundances of acetoclastic and mixotrophic methanogens and Type IIa methanotrophs (dominated by *Methylocystis*), along with soil bulk density and $CO_2$ fluxes. The predictive value of soil respiration, as

reflected by $CO_2$ flux, may demonstrate the importance of overall decomposition rates to $CH_4$ production, consistent with the correlations of carbohydrate-degrading genes (Fig. 2) and taxa (Fig. 4; Fig. S7) with $CH_4$ flux.

SEM and LASSO regression models indicated a strong influence of acetoclastic methanogens on $CH_4$ fluxes (Fig. 6; Fig. S10b); this guild was associated with higher $CH_4$ fluxes elsewhere (98–100) and was previously found to contribute to the majority of $CH_4$ production in our highest $CH_4$ site (101). Although acetoclasts were negatively associated with sulfate-reducing bacteria across all sites, these relationships appeared weakly *positive* when considering only freshwater and oligohaline Delta wetlands (Fig. S5d and f). Acetoclasts, along with the other methanogen guilds, were also positively associated with syntrophic bacteria, particularly in the Delta (Fig. S5e and f). This association could contribute to increased $CH_4$ production under oligohaline conditions if seawater influence were to promote the growth of sulfate-reducing syntrophs, which can produce methanogenic substrates such as acetate and hydrogen (32, 91–94, 102).

Finally, in agreement with our other correlation analyses, integrative SEMs indicated a particular influence of Type IIa methanotrophs ($CH_4$ oxidizers) on $CH_4$ fluxes within high $CH_4$-emitting Delta soils (Fig. 6). These models further indicated effects of the soil structure metrics bulk density and water-filled pore space on $CH_4$ consumption (Fig. 6). Although these models suggest potential metabolic interactions among microbial guilds linked to $CH_4$ fluxes, we acknowledge limitations of this statistical approach. Limited sample size impeded development of more complex and comprehensive models, including prediction of soil $CO_2$ fluxes from taxa or fermentative pathways (32). SEM model stringency also led to the elimination of potentially meaningful factors during model selection due to their covariance with other dominant features. Such factors included hydrogenotrophic methanogens (Fig. 2) which were closely linked to hydrogen production genes (Fig. S5c and d), Type I methanotrophs, and porewater DOC.

## Conclusions

Our study of $CH_4$ fluxes and microbial metabolism across an estuarine wetland salinity gradient found that $CH_4$ fluxes were not a simple function of salinity or sulfate availability. Although $CH_4$ fluxes were generally lower as salinity increased, the highest fluxes were observed in non-tidal restored oligohaline wetlands, consistent with a meta-analysis of tidal marsh $CH_4$ fluxes elsewhere (18). These patterns may suggest that low-level salinity intrusion (i.e., <5 ppt) could increase $CH_4$ flux in tidal freshwater wetlands, while higher levels of salinization (i.e., >5 ppt) might instead decrease $CH_4$ fluxes [up until extreme hypersaline conditions (103)]. Our results further indicated that methanogenesis genes alone did not account for landscape patterns of $CH_4$ fluxes, suggesting mechanisms altering methanogenesis, methanotrophy, nitrogen cycling and ammonium release, and increasing decomposition and syntrophic bacterial populations each could contribute to potential increases in net $CH_4$ flux as a result of salinity intrusion into freshwater soils. Improved understanding of these influences on net $CH_4$ emissions could improve restoration efforts and accounting of blue carbon sequestration in estuarine marshes (47).

We suggest that the potential interactions among salinity, mineral N forms, and methanotrophy may merit further investigation in estuarine wetlands, particularly regarding the response of the dominant Type IIa methanotroph *Methylocystis*, and particularly regarding freshwater to oligohaline salinities. We report some of the highest $CH_4$ emissions in oligohaline sites; this is concerning because seawater intrusion will cause tidal freshwater estuarine wetlands to become oligohaline. More pristine reference sites may have older and more abundant organic matter with higher C:N compared to wetlands impacted by agricultural activity and thus may present different interactions between salinity and $CH_4$. This distinction might be critical for modeling efforts to scale up biogeochemical process interactions in estuarine wetlands (38, 89), given that agricultural nutrient loading impacts the majority of large estuaries globally. This is particularly important in the context of sea level rise and/or drought conditions, both of

which will increasingly cause estuarine wetlands to experience higher salinities than they have historically.

## MATERIALS AND METHODS

### Sample collection

Eleven estuarine wetland complexes (16 total sampling sites, Table S1) were sampled throughout the San Francisco Bay and Delta region, with salinities ranging from 0.4 to 61.5 ppt, between 29 August and 14 October 2013 (Fig. 1a). The salinity gradient encompassed sites with below, equal to, and above seawater salinities of ~35 ppt. This sampling time period corresponds to the window of peak methane flux in this region between July and October (72). All samples from the same sampling site were collected on the same day except for White Slough and wetland complexes with multiple sampling sites (Mayberry, Rush Ranch, and China Camp); at these four locations, $CH_4$ fluxes did not vary significantly by sampling date (analysis of variance, $P > 0.05$). Average daily air temperatures between the sampling dates ranged from 14.6°C to 25.4°C, while 8-cm-deep soil temperature ranged from 12.3°C to 21.0°C, based on data collected at the Mayberry and West Pond sites, which are the only two sites with established flux towers (Ameriflux/FluxNet IDs US-Myb, US-Twi) (72). Site descriptions, soil summary statistics, and sampling details are given in the supplemental text and Table S1. Delta sites range in mean salinity from 0.5 to 3.5 ppt, while Bay sites range in mean salinity from 6.9 to 41.7 ppt.

Wetlands sampled included established reference (undisturbed historic) sites and wetlands restored from former use as agricultural land or as dredged material placement sites. Two of the freshwater and oligohaline restored wetland complexes (West Pond and Mayberry) had been previously characterized for greenhouse gas fluxes (22, 36–38, 104), and one site had been studied for microbe-methane interactions (West Pond) (104). Sampling points were chosen at each site based on dominant vegetation for high and low marsh ecotones (Table S1), and three coring locations (A, B, and C) within plant community type were selected within a 10-m radius at each sampling point. While this sampling scheme likely captures only a fraction of the variation within each site, it enabled us to sample a larger number of sites and span the entire salinity gradient of the estuary at fine increments. Intact soil cores (5-cm diameter, 15 cm deep) were obtained, and following greenhouse gas measurements (see below), they were split into 0- to 5-cm-depth (D1) and 5- to 15-cm-depth (D2) sections. Methane production is known to occur at these depths in wetlands (105, 106). Each section was homogenized and frozen on dry ice in the field, then stored at −80°C prior to DNA extraction and soil geochemical analyses, both of which were conducted on both depths. An additional intact soil core was retrieved adjacent to each of the three DNA soil cores (ca. 50-cm distance) and was transported to the lab at ambient temperature for greenhouse gas analysis. All soils were flooded and surface water was decanted off the cores. Porewater was collected from polyvinyl chloride sampling pipes slotted at 5–10 cm beneath the soil surface, then filtered (0.45 µm) and frozen for subsequent analyses. *In situ* measurements of water pH, temperature, conductivity, dissolved oxygen, reduction-oxidation (redox) potential, and salinity (based on conductivity) were collected from the sediment core holes using a YSI Multi-Parameter Water Quality Sonde (Model 6920-v2; YSI Inc., Yellow Springs, OH, USA).

### Greenhouse gas flux analyses

Intact soil cores (at least three per site) were analyzed for greenhouse gas production ($CH_4$, $CO_2$, and $H_2O$) using a Los Gatos Research greenhouse gas analyzer (GGA; Los Gatos Research, Mountain View, CA, USA), which measures $CO_2$ and $CH_4$ concentrations at 1 Hz. Cores were closed on the bottom with airtight caps and loaded into a 2-L glass Mason jar fitted with airtight tubing to allow continuous gas exchange with the GGA. Fluxes were determined from the linear slope of gas concentrations over the latter of two

consecutive 500-s intervals, with headspace ventilation for 100 s between cycles. Further details of these methods are given in the supplemental text.

## Soil and porewater biogeochemical measurements

Soil carbon content, nutrient concentrations (total N and P, and extractable $NH_4^+$, $NO_3^-$, and $PO_4^{2-}$), pH, and water content were measured for both the 0- to 5-cm-depth and 5- to 15-cm-depth soil samples at the UC Davis Analytical Lab following protocols listed in Table S2. Additionally, soil diethylenetriaminepentaacetic acid extractable metals (Fe, Cu, Mn, and Al) were measured on 5- to 5-cm horizons by the UC Davis Analytical Lab (Table S2). Detailed descriptions of soil chemical methods are also given in Hartman et al. (107). Filtered soil porewater samples were analyzed for total organic carbon (TIC/TOC analyzer) to determine DOC at the Aqueous Chemistry Laboratory at Lawrence Berkeley National Laboratory as described previously (104). Results from biogeochemical measurements are given in Table S3.

## Soil DNA extraction and sequencing

Frozen soil samples were thawed at 4°C and homogenized, and approximately 0.5 g of wet soil sample was removed for DNA extraction from both the 0- to 5-cm (D1) and 5- to 15-cm (D2) soil core strata using the PowerLyzer PowerSoil DNA isolation kit (Mo Bio Laboratories, Inc., Carlsbad, CA, USA). DNA yield was assessed with the Qubit (v.2.0) fluorometer (Invitrogen, Carlsbad, CA, USA). To determine microbial community composition, we amplified the V4 region of the 16S rRNA gene using barcoded primers 515 F (5′-GTGCCAGCMGCCGCGGTAA-3′) and 806 R (5′-GGACTACHVGGGTTCTAAT-3′) established by Caporaso et al. (108). Amplicon sequencing was performed following the JGI's standard protocols (detailed in the supplemental text), where 16S rRNA gene amplicons were diluted to 10 nM, quantified by qPCR, and sequenced on the Illumina MiSeq platform (2 × 300 bp, Reagent Kit v.3; Illumina Inc., Carlsbad, CA, USA).

16S rRNA gene amplicon sequences were analyzed using the iTagger (v.1.1) pipeline (109), which removed Illumina adapters and PhiX sequences, performed paired-end read assembly, read quality filtering, and chimera checking, and clustered reads into operational taxonomic units (OTUs) at 97% similarity. Taxonomic classification of OTUs was achieved using the "assignTaxonomy" function in the *dada2* R package (110) and the SILVA (v.138.1) reference database (111). Microbial sequence reads were further aggregated into functional guilds using taxonomic assignments of 16S rRNA gene reads for groups including acetoclastic, hydrogenotrophic, mixotrophic, and methyl-reducing methanogens, groups of microbes oxidizing $CH_4$, ammonia, and nitrite, and sulfate-reducing bacteria. These assignments were based on monophyletic functional groups derived from taxonomic patterns in the literature, described in detail in the supplemental text, along with further details of amplicon sequence data processing methods. Mixotrophic methanogens are taxa that are capable of performing at least two different methanogenesis pathways and include Methanosarcinaceae (contain taxa that can perform one or more of all four pathways) and Methanobacteriaceae (contain taxa that can perform hydrogenotrophic methanogenesis or methyl-reducing methanogenesis) (44, 112, 113). Taxa in the Nitrospirota phylum were all considered to be nitrite-oxidizing bacteria, although some taxa (in the *Nitrospira* genus) have been recently discovered to perform complete ammonia oxidation (114

Soil metagenomic shotgun sequence data were obtained using a 96-well plate-based DNA library preparation (detailed in the supplemental text) run on an Illumina HiSeq2500 sequencer using HiSeq TruSeq SBS (v.4) sequencing kits (2 × 150 or 2 × 250 run mode) at the Joint Genome Institute. Overall, shotgun sequencing libraries yielded ~5.3 Gbp per sample after contaminant and quality filtering. Unassembled FASTQ-formatted sequencing read data from each sample were submitted to the MG-RAST metagenome annotation server (60, 115), with details of the underlying bioinformatics algorithms described in the supplemental text. MG-RAST and Genomes OnLine Database accession numbers for the metagenomes are presented in Table

S3. Counts of functional annotations organized by the Kyoto Encyclopedia of Genes and Genomes (KEGG) Ortholog (KO) (61) were downloaded for each sample from the MG-RAST application programming interface (API) (60) using a custom Python script which merged annotations into a single table of counts for each KO for each sample.

Further annotation of unassembled shotgun sequence reads for specified microbial functional guilds was accomplished using the TreeSAPP (v.0.6.0) pipeline (116), which identifies open reading frames(ORFs), annotates gene function, and assigns taxonomy based on phylogenetic placement relative to reference sequences. FunGene (v.9.5) was used to download reference sequences. This approach was applied to methanogens (*mcrABG*), sulfate reducers (*dsrAB*), ammonia and $CH_4$ oxidizers (*amo/pmoABC*), and nitrite oxidizers (*nxrAB*), along with single-copy marker genes for DNA replication (*recA*, *rpoB*, and *RPS3A*), as detailed in the supplemental text. Parameters for the TreeSAPP *assign* command are stated in the supplemental text; the TreeSAPP "create" command for identifying sequencing reads homologous to each reference package used the default parameters. This analysis was particularly important for *pmoA-amoA*, which was not annotated by MG-RAST. While none of the other key carbon, nitrogen, phosphorus, or sulfur genes were missed by MG-RAST, it is possible that other genes were not annotated. TreeSAPP also performed better, in terms of correlations with 16S guild relative abundances, than *in silico* PCR, which was tested for *mcrA* and *pmoA* genes (Fig. S13) using the mlas-mod-F/mcrA-rev-R primers (117) and the A189-mb661 primers (118), respectively, implemented with the pcr-seqs command in the *mothur* software (119).

## Statistical analyses

Statistical analyses and data visualizations were conducted using custom scripts developed in Python and R (120), which are publicly available on GitHub (https://github.com/cliffbueno/SF_microbe_methane). Greenhouse gas fluxes and soil chemical data were log-transformed prior to regression analysis (linear, segmented linear, and polynomial) and visualization using ggplot2 (121) in R. Segmented linear regressions were performed with the "segmented" R package (122). Effects of depth and the nested categorical variables location and wetland status were tested with LME models with the R package "nlme" (123). Gene relative abundance data obtained from MG-RAST were normalized using the "DESeq2" package in R (124), and regressions of log2 transformed DESeq2-normalized counts with environmental factors were compared while controlling false discovery rate of <0.05. Heatmap summary plots of gene-environment relationships were generated using the Seaborn library in Python. LASSO multivariate selection models (125) for predicting $CH_4$ fluxes from sets of soil chemical measurements, genes, and taxonomy-based functional guilds were implemented with the Scikit-learn Python package.

16S rRNA counts of OTUs were also normalized with the DESeq2 package in R to analyze taxonomic relative abundance, while center-log-ratio transformation implemented in the "zCompositions" R package (126) was used to analyze composition. Microbial community composition was analyzed with an Aitchison distance matrix calculated with the "compositions" R package (127), and PERMANOVA test implemented with the "vegan" R package (128), and visualized with principal component analysis. Different taxonomic levels were tested for correlations with methane. Finally, directional interrelationships between $CH_4$ fluxes, decomposition, and microbial guilds were evaluated by structural equation modeling using the R package "lavaan" (129). Models to test mechanistic hypotheses about environmental and microbial drivers of methane flux were developed with methane generation and methane oxidation as composite variables that drive methane flux. SEMs were run for the entire data set as well as the Delta sites alone.

## ACKNOWLEDGMENTS

We gratefully acknowledge the assistance of several parties in selecting and accessing wetland sampling sites, including Bryan Brock (California Department of Water Resources), Matt Ferner (San Francisco State University), John Callaway (University of San Francisco), Lisamarie Windham-Myers (US Geological Survey), and Larry Wykoff (California Fish and Wildlife Service), along with the US National Estuarine Research Reserve program.

This project was funded by the DOE Early Career Research Program (grant no. KP/CH57/1). The work conducted by the U.S. Department of Energy Joint Genome Institute, a Department of Energy Office of Science user facility, is supported by the Office of Science of the U.S. Department of Energy under Contract No. DE-AC02-05CH11231.

## AUTHOR AFFILIATIONS

[1]DOE Joint Genome Institute, Berkeley, California, USA

[2]Department of Microbiology and Immunology, University of British Columbia, Vancouver, British Columbia, Canada

[3]Department of Environmental Science, Policy, and Management, University of California, Berkeley, California, USA

[4]Environmental Genomics and Systems Biology Division, Lawrence Berkeley National Laboratory, Berkeley, California, USA

## PRESENT ADDRESS

Susanna M. Theroux, Southern California Coastal Water Research Project, Costa Mesa, California, USA

## AUTHOR ORCIDs

Clifton P. Bueno de Mesquita http://orcid.org/0000-0002-2565-7100
Susannah G. Tringe http://orcid.org/0000-0001-6479-8427

## FUNDING

| Funder | Grant(s) | Author(s) |
| --- | --- | --- |
| U.S. Department of Energy (DOE) | KP/CH57/1 | Susannah G. Tringe |
| U.S. Department of Energy (DOE) | DE-AC02-05CH11231 | Susannah G. Tringe |
| | | Wyatt H. Hartman |
| | | Clifton P. Bueno de Mesquita |
| | | Susanna M. Theroux |

## AUTHOR CONTRIBUTIONS

Wyatt H. Hartman, Conceptualization, Data curation, Formal analysis, Investigation, Methodology, Software, Visualization, Writing – original draft | Clifton P. Bueno de Mesquita, Data curation, Formal analysis, Investigation, Visualization, Writing – review and editing | Susanna M. Theroux, Conceptualization, Data curation, Investigation, Methodology, Writing – review and editing | Connor Morgan-Lang, Data curation, Formal analysis, Investigation, Methodology, Software, Writing – review and editing | Dennis D. Baldocchi, Data curation, Methodology, Validation, Writing – review and editing | Susannah G. Tringe, Conceptualization, Investigation, Methodology, Project administration, Resources, Supervision, Writing – review and editing

## DATA AVAILABILITY

Raw and processed metagenomic and 16S rRNA gene data and metadata are publicly available; Genomes OnLine Database/Integrated Microbial Genomes, National Center for Biotechnology Information, and Metagenomics Rapid Annotation using Subsystems Technology accession information is presented in Table S3. Biogeochemical data are presented in Table S4. All supplementary tables are also publicly available on Figshare (10.6084/m9.figshare.24808383). All analysis scripts are available on GitHub.

## ADDITIONAL FILES

The following material is available online.

### Supplemental Material

**Supplemental Figures (mSystems00936-23-s0001.docx).** Figures S1 to S13.
**Supplemental Text (mSystems00936-23-s0002.docx).** More methodological details.
**Supplemental Tables (mSystems00936-23-s0003.xlsx).** Tables S1 to S8.

### Open Peer Review

**PEER REVIEW HISTORY (review-history.pdf).** An accounting of the reviewer comments and feedback.

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
