## [Reviewer comments · mSystems]

Multiple microbial guilds mediate soil methane cycling along a wetland salinity gradient

Wyatt Hartman, Clifton Bueno de Mesquita, Susanna Theroux, Connor Morgan-Lang, Dennis Baldocchi, and Susannah Tringe

Corresponding Author(s): Susannah Tringe, E O Lawrence Berkeley National Laboratory

Review Timeline:

Submission Date:	September 11, 2023
Editorial Decision:	October 8, 2023
Revision Received:	November 16, 2023
Editorial Decision:	November 22, 2023
Revision Received:	November 27, 2023
Accepted:	November 29, 2023

Editor: Marcela Hernandez

Reviewer(s): The reviewers have opted to remain anonymous.

Transaction Report:

DOI: <https://doi.org/10.1128/msystems.00936-23>

October 8, 2023

Dr. Susannah Green Tringe
E O Lawrence Berkeley National Laboratory
Environmental Genomics and Systems Biology
1 Cyclotron Rd
Berkeley, CA 94720

Re: mSystems00936-23 (Multiple microbial guilds mediate soil methane cycling along a wetland salinity gradient)

Dear Dr. Susannah Green Tringe:

Thank you for submitting your manuscript to mSystems. We have completed our review and I am pleased to inform you that, in principle, we expect to accept it for publication in mSystems. However, acceptance will not be final until you have adequately addressed the reviewer comments.

Even though Reviewer 1 appreciated the authors' efforts to clarify issues with metabolic gene annotation, Reviewer 2 has some important further comments that I would like the authors to discuss. Reviewer 2 is suggesting some complementary, relatively fast, analyses with qPCR assays targeting specific genes related to methane processes. Additionally, they express concern that certain genes, like *pmoA*, were not detected in the metagenome analysis despite the presence of proteobacterial-affiliated methanotrophs in 16S rRNA gene sequencing. This raises doubts about potential missed genes in the metagenome analysis, and they recommend addressing this limitation in the discussion.

Preparing Revision Guidelines

Please return the manuscript within 60 days; if you cannot complete the modification within this time period, please contact me. If you do not wish to modify the manuscript and prefer to submit it to another journal, please notify me of your decision immediately so that the manuscript may be formally withdrawn from consideration by mSystems.

Sincerely,

Marcela Hernandez

Editor, mSystems

Journals Department
Reviewer comments:

Reviewer #1 (Comments for the Author):

Thanks to the authors for changes to the text to clarify issues with metabolic gene annotation. The TREESAPP based analysis adds value. Can the authors include parameters such as sequence length cut-off? Based on the methods/supplementary text the metagenomes were not assembled - more information on the specific parameters to assign taxa via TREESAPP will benefit the readers. I have no other major concern with the manuscript

Reviewer #2 (Comments for the Author):

Comments to authors

Hartman et al. determined the CO₂ and CH₄ fluxes of soil cores sampled from sites with a range of salinity levels. The flux measurements were complemented with sequencing analyses (16S rRNA gene-targeted amplicon sequencing and shotgun sequencing for metagenome analysis) to detect trends in taxonomic and functional patterns along the salinity gradient. I did not review the previous version of this manuscript. I feel that the authors have adequately responded to the reviewers' comments, and the research topic is relevant. The manuscript is descriptive, and well-written.

General comments

1. Although methane flux was measured, the potential (rates) for methane production and oxidation remain to be determined, and largely depended on the metagenome analysis (qualitative results) to provide insights into these processes (correlating the relative abundances of specific genes to processes across sites). Instead of targeting the gene transcript (which will likely yield more conclusive interpretation of the results), the study is DNA-based. As such, this study tends to be speculative.

Provided DNA extracts are still available, qPCR assays targeting specific genes and/or gene transcripts of interest (*mcrA*, *pmoA*, and genes involved in nitrification/denitrification, sulphate reduction, cellulose degradation) can be performed to complement the metagenome analysis. Note that qPCR assays for these genes are available.

2. Echoing the point raised by the previous reviews, it is disturbing that although proteobacterial-affiliated methanotrophs (proven to harbour the *pmoA* gene) were detected in the 16S rRNA gene sequencing analysis, the *pmoA* gene escaped detection in the metagenome analysis. This concern has been addressed by the authors but raised doubts whether other genes may have been missed in the metagenome analysis. Perhaps, include this limitation in the discussion.

Specific comments

1. Lines 125-130, and figure 1d,e,f: Are there (significant) differences in the soil chemistry between the 0-5 cm and 5-15cm soil depth? The different depth layers were not differentiated in the figure.

2. Lines 185, 241, 664, 652, 413-418, Figure 3 legend, and elsewhere throughout the manuscript: The sequencing analyses (16S rRNA gene amplicons and HiSeq) provide "relative" gene abundances and should be treated/stated as such.

3. Line 262: Please double check; *Methylotenera* is a non-methanotrophic methylotroph (not a methanotroph).

4. Line 276, 278, and elsewhere: I feel that terms like "considerably higher" are not meaningful. Are the differences/correlations statistically significant (i.e., "significantly", rather than "considerably" higher)?

5. Line 393: The bio-availability (rather than the availability) of ammonium, which can be estimated from the soluble fraction, would be more relevant (e.g., please see van Dijk et al., 2021, *Biology and Fertility of Soils* 57: 873-880) for the methanotrophs

and methanotrophic activity.

6. Line 527: How does the salinity range (0.4 - 61.5 ppt) compare to seawater salinity level?

7. Line 578: What was the rationale of determining the extractable metals only at the deeper depth layer (5-15 cm)?

Response to Reviewers

We thank the editor and two reviewers for their feedback. Please see our responses in bold. Line numbers refer to the line numbers in the document with tracked changes.

Editor:

Dear Dr. Susannah Green Tringe:

Thank you for submitting your manuscript to mSystems. We have completed our review and I am pleased to inform you that, in principle, we expect to accept it for publication in mSystems. However, acceptance will not be final until you have adequately addressed the reviewer comments.

Thank you for handling our manuscript and for this decision. We hope we have adequately addressed the comments below.

Even though Reviewer 1 appreciated the authors' efforts to clarify issues with metabolic gene annotation, Reviewer 2 has some important further comments that I would like the authors to discuss. Reviewer 2 is suggesting some complementary, relatively fast, analyses with qPCR assays targeting specific genes related to methane processes. Additionally, they express concern that certain genes, like *pmoA*, were not detected in the metagenome analysis despite the presence of proteobacterial-affiliated methanotrophs in 16S rRNA gene sequencing. This raises doubts about potential missed genes in the metagenome analysis, and they recommend addressing this limitation in the discussion.

We agree the discrepancies in metagenome annotation are concerning, which is why we compared multiple methods of annotating critical genes. The MG-RAST analysis was a critical first step, as it was the only software that would accept unassembled reads for annotation and we didn't want to bias the analysis by focusing solely on the assembled contigs, or annotating only a preselected set of functional genes. To more accurately quantify functional genes of interest in the metagenome data, we used TreeSAPP, which produced results much more consistent with the 16S data (shown in Figure S8). We believe these TreeSAPP analyses are as quantitative and specific, if not more so, than qPCR data would be as we discuss further in our response to Reviewer 2.

Reviewer #1 (Comments for the Author):

Thanks to the authors for changes to the text to clarify issues with metabolic gene annotation. The TREESAPP based analysis adds value. Can the authors include parameters such as sequence length cut-off? Based on the methods/supplementary text the metagenomes were not assembled - more information on the specific parameters to assign taxa via TREESAPP will benefit the readers. I have no other major concern with the manuscript

Thank you for reviewing our manuscript again, and for these additional suggestions.

We are pleased that our revisions adequately addressed your previous concerns. We have now clarified that unassembled metagenomes were submitted to MG-RAST (line 633) and used for TreeSAPP analysis (line 640).

We have now added brief details on the TreeSAPP parameters to lines 647-649, which state: “Parameters for the TreeSAPP *assign* command are stated in the Supplemental text; TreeSAPP *create* command for identifying sequencing reads homologous to each reference package used the default parameters.”

Additional details can now be found in the TreeSAPP section (Section VII) of the supplementary text, which now states: “The TreeSAPP commands used to create reference packages (with ``treesapp create``) included the following arguments: `--trim_align` to trim the multiple sequence alignment with BMGE prior to inferring a reference phylogeny; `--fast` to use FastTree for inferring the phylogeny; `--cluster` and `-p <sequence identity>` to dereplicate the input sequences, reducing the size of the reference phylogeny and thereby increasing phylogenetic placement speed while maintaining diversity; `--screen 'Bacteria,Archaea'` to remove sequences derived from Eukaryotes; `--min_taxonomic_rank c` to ensure all reference sequences have a taxonomic lineage resolved to at least the class rank; `--profile <hmm file>` to use homologous sequences identified by hidden Markov model (HMM) alignment in cases where curation was deemed insufficient or protein families contained accessory domains that interfered with alignment and phylogenetic inference. This was the case for NxrA (COG5013), RecA/RadA (COG0468), RpoB (COG0085), and DsrAB (COG2221), for which HMMs were downloaded from EggNOG v4.5. An HMM profile wasn't necessary to help curate McrA and XmoA sequences as these were manually reviewed. The ``treesapp assign`` commands for identifying sequencing reads homologous to each reference package used the default parameters. The parameter defaults related to sequence processing and filtering required HMM sequence alignments to have a maximum E-value of 1E-5 and a minimum bit score of 20, and placement likelihoods (computed by RAxML) needed to exceed 0.1. No multiple sequence alignment trimming was done with BMGE prior to sequence placement.”

Reviewer #2 (Comments for the Author):

Comments to authors

Hartman et al. determined the CO₂ and CH₄ fluxes of soil cores sampled from sites with a range of salinity levels. The flux measurements were complemented with sequencing analyses (16S rRNA gene-targeted amplicon sequencing and shotgun sequencing for metagenome analysis) to detect trends in taxonomic and functional patterns along the salinity gradient. I did not review the previous version of this manuscript. I feel that the authors have adequately responded to the reviewers' comments, and the research topic is relevant. The manuscript is descriptive, and well-written.

Thank you for your thoughtful review and for your general and specific comments, which we address below.

General comments

1. Although methane flux was measured, the potential (rates) for methane production and oxidation remain to be determined, and largely depended on the metagenome analysis (qualitative results) to provide insights into these processes (correlating the relative abundances of specific genes to processes across sites). Instead of targeting the gene transcript (which will likely yield more conclusive interpretation of the results), the study is DNA-based. As such, this study tends to be speculative.

Provided DNA extracts are still available, qPCR assays targeting specific genes and/or gene transcripts of interest (*mcrA*, *pmoA*, and genes involved in nitrification/denitrification, sulphate reduction, cellulose degradation) can be performed to complement the metagenome analysis. Note that qPCR assays for these genes are available.

We agree with the reviewer about the limitations of the DNA-based approach, and this is pointed out in the text (lines 371-374). While qPCR assays could be informative regarding gene abundance, comprehensive analysis of genes involved in all the processes would be a significant undertaking, and since the DNA is over a decade old it would likely be too degraded for quantitative analysis. We believe the homology-based TreeSAPP approach is equally quantitative as compared with the primer-binding-based qPCR approach, and to explore this we performed *in silico* PCR analysis to essentially simulate the qPCR results. This was implemented with the *mothur* *pcr-seqs* command on unassembled reads for the forward *pmoA* primer A189 and the reverse *pmoA* primer mb661 (Bourne et al. 2001), as well as the forward *mcrA* primer mlas-mod-F and the reverse *mcrA* primer mcrA-rev-R (Angel et al. 2012). We found that TreeSAPP counts are more highly correlated with the

16S guild assignments than isPCR primer match counts, particularly for *pmoA*, suggesting the TreeSAPP method is effectively capturing the relevant genes. Those results are presented in a new supplemental figure (Figure S13), which is now referred to in lines 653-657.

References:

1. Bourne DG, McDonald IR, Murrell JC. Comparison of *pmoA* PCR primer sets as tools for investigating methanotroph diversity in three Danish soils. *Appl Environ Microbiol* 2001; 67: 3802–3809.
2. Angel R, Claus P, Conrad R. Methanogenic archaea are globally ubiquitous in aerated soils and become active under wet anoxic conditions. *ISME J* 2012; 6: 847–862.

2. Echoing the point raised by the previous reviews, it is disturbing that although proteobacterial-affiliated methanotrophs (proven to harbour the *pmoA* gene) were detected in the 16S rRNA gene sequencing analysis, the *pmoA* gene escaped detection in the metagenome analysis. This concern has been addressed by the authors but raised doubts whether other genes may have been missed in the metagenome analysis. Perhaps, include this limitation in the discussion.

To be clear, the *pmoA* gene was detected in metagenome analysis using TreeSAPP, just not in the MG-RAST analysis. None of the other specific genes of interest were missed by MG-RAST, but we have added a statement warning readers that potentially some other non-CNPS genes could have been missed (lines 650-653). The line states: “While none of the other key carbon, nitrogen, phosphorus, or sulfur genes were missed by MG-RAST, it is possible that other genes were not annotated.” This would not affect the overall conclusions of our manuscript, however, because we did not present results about the total number of genes or the or overall composition of genes. We used a targeted approach to only present and discuss the relative abundance of relevant carbon, nitrogen, phosphorus, and sulfur cycling genes.

Specific comments

1. Lines 125-130, and figure 1d,e,f: Are there (significant) differences in the soil chemistry between the 0-5 cm and 5-15cm soil depth? The different depth layers were not differentiated in the figure.

We added the sentence (lines 130-131): “Soil C, N:P, and ammonium were not significantly affected by depth (LME, $p > 0.05$).”

2. Lines 185, 241, 664, 652, 413-418, Figure 3 legend, and elsewhere throughout the manuscript: The sequencing analyses (16S rRNA gene amplicons and HiSeq) provide "relative" gene abundances and should be treated/stated as such.

We updated all of the text accordingly.

3. Line 262: Please double check; Methylothera is a non-methanotrophic methylotroph (not a methanotroph).

Thank you for catching this. We updated the text and Figures 5, S5, S8, S10a, S11, S12 accordingly.

4. Line 276, 278, and elsewhere: I feel that terms like "considerably higher" are not meaningful. Are the differences/correlations statistically significant (i.e., "significantly", rather than "considerably" higher)?

We changed all instances of “considerably” to “significantly”, and confirmed those statements statistically.

5. Line 393: The bio-availability (rather than the availability) of ammonium, which can be estimated from the soluble fraction, would be more relevant (e.g., please see van Dijk et al., 2021, *Biology and Fertility of Soils* 57: 873-880) for the methanotrophs and methanotrophic activity.

Thank you for suggesting this very relevant reference. We will keep this in mind for our future work. Unfortunately, we did not measure the soluble fraction in this project, but we have added this point (and reference) to the discussion. Please see lines 408-409 which read: “The soluble (bioavailable) fraction of ammonium is particularly important for predicting the effects of ammonium on methane oxidation (85).”

6. Line 527: How does the salinity range (0.4 - 61.5 ppt) compare to seawater salinity level?

We added the following sentence (lines 539-540): “The salinity gradient encompassed sites with salinities below, equal to, and above seawater salinities of ~35 ppt.”

7. Line 578: What was the rationale of determining the extractable metals only at the deeper depth layer (5-15 cm)?

Due to limited funds, we could only run metal analyses on one depth and we opted for the deeper depth where we expected more methanogenesis to occur. There was also more soil material available for analysis at the deeper depth as it contained an extra 5 cm of depth compared to the upper layer (i.e., 5-15 cm segment vs. 0-5 cm segment).

Re: mSystems00936-23R1 (Multiple microbial guilds mediate soil methane cycling along a wetland salinity gradient)

Dear Dr. Susannah Green Tringe:

Please, send a new rebuttal letter in which the line numbers in the responses match the clean revised version.

Revision Guidelines

Sincerely,
Marcela Hernandez
Editor
mSystems

Response to Reviewers

We thank the editor and two reviewers for their feedback. Please see our responses in bold. Line numbers refer to the line numbers in the clean manuscript document.

Editor:

Dear Dr. Susannah Green Tringe:

Thank you for submitting your manuscript to mSystems. We have completed our review and I am pleased to inform you that, in principle, we expect to accept it for publication in mSystems. However, acceptance will not be final until you have adequately addressed the reviewer comments.

Thank you for handling our manuscript and for this decision. We hope we have adequately addressed the comments below.

Even though Reviewer 1 appreciated the authors' efforts to clarify issues with metabolic gene annotation, Reviewer 2 has some important further comments that I would like the authors to discuss. Reviewer 2 is suggesting some complementary, relatively fast, analyses with qPCR assays targeting specific genes related to methane processes. Additionally, they express concern that certain genes, like *pmoA*, were not detected in the metagenome analysis despite the presence of proteobacterial-affiliated methanotrophs in 16S rRNA gene sequencing. This raises doubts about potential missed genes in the metagenome analysis, and they recommend addressing this limitation in the discussion.

We agree the discrepancies in metagenome annotation are concerning, which is why we compared multiple methods of annotating critical genes. The MG-RAST analysis was a critical first step, as it was the only software that would accept unassembled reads for annotation and we didn't want to bias the analysis by focusing solely on the assembled contigs, or annotating only a preselected set of functional genes. To more accurately quantify functional genes of interest in the metagenome data, we used TreeSAPP, which produced results much more consistent with the 16S data (shown in Figure S8). We believe these TreeSAPP analyses are as quantitative and specific, if not more so, than qPCR data would be as we discuss further in our response to Reviewer 2.

Reviewer #1 (Comments for the Author):

Thanks to the authors for changes to the text to clarify issues with metabolic gene annotation. The TREESAPP based analysis adds value. Can the authors include parameters such as sequence length cut-off? Based on the methods/supplementary text the metagenomes were not assembled - more information on the specific parameters to assign taxa via TREESAPP will benefit the readers. I have no other major concern with the manuscript

Thank you for reviewing our manuscript again, and for these additional suggestions.

We are pleased that our revisions adequately addressed your previous concerns. We have now clarified that unassembled metagenomes were submitted to MG-RAST (line 625) and used for TreeSAPP analysis (line 632).

We have now added brief details on the TreeSAPP parameters to lines 639-641, which state: “Parameters for the TreeSAPP *assign* command are stated in the Supplemental text; TreeSAPP *create* command for identifying sequencing reads homologous to each reference package used the default parameters.”

Additional details can now be found in the TreeSAPP section (Section VII) of the supplementary text, which now states: “The TreeSAPP commands used to create reference packages (with ``treesapp create``) included the following arguments: `--trim_align` to trim the multiple sequence alignment with BMGE prior to inferring a reference phylogeny; `--fast` to use FastTree for inferring the phylogeny; `--cluster` and `-p <sequence identity>` to dereplicate the input sequences, reducing the size of the reference phylogeny and thereby increasing phylogenetic placement speed while maintaining diversity; `--screen 'Bacteria,Archaea'` to remove sequences derived from Eukaryotes; `--min_taxonomic_rank c` to ensure all reference sequences have a taxonomic lineage resolved to at least the class rank; `--profile <hmm file>` to use homologous sequences identified by hidden Markov model (HMM) alignment in cases where curation was deemed insufficient or protein families contained accessory domains that interfered with alignment and phylogenetic inference. This was the case for NxrA (COG5013), RecA/RadA (COG0468), RpoB (COG0085), and DsrAB (COG2221), for which HMMs were downloaded from EggNOG v4.5. An HMM profile wasn't necessary to help curate McrA and XmoA sequences as these were manually reviewed. The ``treesapp assign`` commands for identifying sequencing reads homologous to each reference package used the default parameters. The parameter defaults related to sequence processing and filtering required HMM sequence alignments to have a maximum E-value of 1E-5 and a minimum bit score of 20, and placement likelihoods (computed by RAxML) needed to exceed 0.1. No multiple sequence alignment trimming was done with BMGE prior to sequence placement.”

Reviewer #2 (Comments for the Author):

Comments to authors

Hartman et al. determined the CO₂ and CH₄ fluxes of soil cores sampled from sites with a range of salinity levels. The flux measurements were complemented with sequencing analyses (16S rRNA gene-targeted amplicon sequencing and shotgun sequencing for metagenome analysis) to detect trends in taxonomic and functional patterns along the salinity gradient. I did not review the previous version of this manuscript. I feel that the authors have adequately responded to the reviewers' comments, and the research topic is relevant. The manuscript is descriptive, and well-written.

Thank you for your thoughtful review and for your general and specific comments, which we address below.

General comments

1. Although methane flux was measured, the potential (rates) for methane production and oxidation remain to be determined, and largely depended on the metagenome analysis (qualitative results) to provide insights into these processes (correlating the relative abundances of specific genes to processes across sites). Instead of targeting the gene transcript (which will likely yield more conclusive interpretation of the results), the study is DNA-based. As such, this study tends to be speculative.

Provided DNA extracts are still available, qPCR assays targeting specific genes and/or gene transcripts of interest (*mcrA*, *pmoA*, and genes involved in nitrification/denitrification, sulphate reduction, cellulose degradation) can be performed to complement the metagenome analysis. Note that qPCR assays for these genes are available.

We agree with the reviewer about the limitations of the DNA-based approach, and this is pointed out in the text (lines 364-367). While qPCR assays could be informative regarding gene abundance, comprehensive analysis of genes involved in all the processes would be a significant undertaking, and since the DNA is over a decade old it would likely be too degraded for quantitative analysis. We believe the homology-based TreeSAPP approach is equally quantitative as compared with the primer-binding-based qPCR approach, and to explore this we performed *in silico* PCR analysis to essentially simulate the qPCR results. This was implemented with the *mothur* *pcr-seqs* command on unassembled reads for the forward *pmoA* primer A189 and the reverse *pmoA* primer mb661 (Bourne et al. 2001), as well as the forward *mcrA* primer mlas-mod-F and the reverse *mcrA* primer mcrA-rev-R (Angel et al. 2012). We found that TreeSAPP counts are more highly correlated with the

16S guild assignments than isPCR primer match counts, particularly for *pmoA*, suggesting the TreeSAPP method is effectively capturing the relevant genes. Those results are presented in a new supplemental figure (Figure S13), which is now referred to in lines 644-648.

References:

1. Bourne DG, McDonald IR, Murrell JC. Comparison of *pmoA* PCR primer sets as tools for investigating methanotroph diversity in three Danish soils. *Appl Environ Microbiol* 2001; 67: 3802–3809.
2. Angel R, Claus P, Conrad R. Methanogenic archaea are globally ubiquitous in aerated soils and become active under wet anoxic conditions. *ISME J* 2012; 6: 847–862.

2. Echoing the point raised by the previous reviews, it is disturbing that although proteobacterial-affiliated methanotrophs (proven to harbour the *pmoA* gene) were detected in the 16S rRNA gene sequencing analysis, the *pmoA* gene escaped detection in the metagenome analysis. This concern has been addressed by the authors but raised doubts whether other genes may have been missed in the metagenome analysis. Perhaps, include this limitation in the discussion.

To be clear, the *pmoA* gene was detected in metagenome analysis using TreeSAPP, just not in the MG-RAST analysis. None of the other specific genes of interest were missed by MG-RAST, but we have added a statement warning readers that potentially some other non-CNPS genes could have been missed (lines 642-644). The line states: “While none of the other key carbon, nitrogen, phosphorus, or sulfur genes were missed by MG-RAST, it is possible that other genes were not annotated.” This would not affect the overall conclusions of our manuscript, however, because we did not present results about the total number of genes or the or overall composition of genes. We used a targeted approach to only present and discuss the relative abundance of relevant carbon, nitrogen, phosphorus, and sulfur cycling genes.

Specific comments

1. Lines 125-130, and figure 1d,e,f: Are there (significant) differences in the soil chemistry between the 0-5 cm and 5-15cm soil depth? The different depth layers were not differentiated in the figure.

We added the sentence (lines 130-131): “Soil C, N:P, and ammonium were not significantly affected by depth (LME, $p > 0.05$).”

2. Lines 185, 241, 664, 652, 413-418, Figure 3 legend, and elsewhere throughout the manuscript: The sequencing analyses (16S rRNA gene amplicons and HiSeq) provide "relative" gene abundances and should be treated/stated as such.

We updated all of the text accordingly.

3. Line 262: Please double check; Methylothera is a non-methanotrophic methylotroph (not a methanotroph).

Thank you for catching this. We updated the text and Figures 5, S5, S8, S10a, S11, S12 accordingly.

4. Line 276, 278, and elsewhere: I feel that terms like "considerably higher" are not meaningful. Are the differences/correlations statistically significant (i.e., "significantly", rather than "considerably" higher)?

We changed all instances of “considerably” to “significantly”, and confirmed those statements statistically.

5. Line 393: The bio-availability (rather than the availability) of ammonium, which can be estimated from the soluble fraction, would be more relevant (e.g., please see van Dijk et al., 2021, *Biology and Fertility of Soils* 57: 873-880) for the methanotrophs and methanotrophic activity.

Thank you for suggesting this very relevant reference. We will keep this in mind for our future work. Unfortunately, we did not measure the soluble fraction in this project, but we have added this point (and reference) to the discussion. Please see lines 401-402 which read: “The soluble (bioavailable) fraction of ammonium is particularly important for predicting the effects of ammonium on methane oxidation (85).”

6. Line 527: How does the salinity range (0.4 - 61.5 ppt) compare to seawater salinity level?

We added the following sentence (lines 531-532): “The salinity gradient encompassed sites with salinities below, equal to, and above seawater salinities of ~35 ppt.”

7. Line 578: What was the rationale of determining the extractable metals only at the deeper depth layer (5-15 cm)?

Due to limited funds, we could only run metal analyses on one depth and we opted for the deeper depth where we expected more methanogenesis to occur. There was also more soil material available for analysis at the deeper depth as it contained an extra 5 cm of depth compared to the upper layer (i.e., 5-15 cm segment vs. 0-5 cm segment).

Re: mSystems00936-23R2 (Multiple microbial guilds mediate soil methane cycling along a wetland salinity gradient)

Dear Dr. Susannah Green Tringe:

Your manuscript has been accepted, and I am forwarding it to the ASM production staff for publication. Your paper will first be checked to make sure all elements meet the technical requirements. ASM staff will contact you if anything needs to be revised before copyediting and production can begin. Otherwise, you will be notified when your proofs are ready to be viewed.

Featured Image Submissions: If you would like to submit a potential Featured Image, please email a file and a short legend to mSystems@asmusa.org. Please note that we can only consider images that (i) the authors created or own and (ii) have not been previously published. By submitting, you agree that the image can be used under the same terms as the published article. File requirements: square dimensions (4" x 4"), 300 dpi resolution, RGB colorspace, TIF file format.

Sincerely,
Marcela Hernandez
Editor
mSystems